# Why Has Breast Cancer Screening Failed to Decrease the Incidence of de Novo Stage IV Disease?

**DOI:** 10.3390/cancers11040500

**Published:** 2019-04-08

**Authors:** Danielle R. Heller, Alexander S. Chiu, Kaitlin Farrell, Brigid K. Killelea, Donald R. Lannin

**Affiliations:** 1Department of Surgery, Yale School of Medicine, New Haven, CT 06520, USA; d.heller@yale.edu (D.R.H.); alex.chiu@yale.edu (A.S.C.); 2The Breast Center/Section of Surgical Oncology, Department of Surgery, Yale School of Medicine, New Haven, CT 06520, USA; kaitlinfarrell10@gmail.com (K.F.); Brigid.killelea@yale.edu (B.K.K.)

**Keywords:** breast cancer, stage IV, incidence, tumor biology, NCDB, SEER

## Abstract

*Background*: Despite screening mammography, the incidence of Stage IV breast cancer (BC) at diagnosis has not decreased over the past four decades. We previously found that many BCs are small due to favorable biology rather than early detection. This study compared the biology of Stage IV cancers with that of small cancers typically found by screening. *Methods*: Trends in the incidence of localized, regional, and distant female BC were compared using SEER*Stat. The National Cancer Database (NCDB) was then queried for invasive cancers from 2010 to 2015, and patient/disease variables were compared across stages. Biological variables including estrogen receptor (ER), progesterone receptor (PR), human epidermal growth factor receptor 2 (Her2), grade, and lymphovascular invasion were sorted into 48 combinations, from which three biological subtypes emerged: indolent, intermediate, and aggressive. The distributions of the subtypes were compared across disease stages. Multivariable regression assessed the association between Stage IV disease and biology. *Results*: SEER*Stat confirmed that the incidence of distant BC increased between 1973 and 2015 (annual percent change [APC] = 0.46). NCDB data on roughly 993,000 individuals showed that Stage IV disease at presentation is more common in young, black, uninsured women with low income/education and large, biologically aggressive tumors. The distribution of tumor biology varied by stage, with Stage IV disease including 37.6% aggressive and 6.0% indolent tumors, versus sub-centimeter Stage I disease that included 5.1% aggressive and 40.6% indolent tumors (*p* < 0.001). The odds of Stage IV disease presentation more than tripled for patients with aggressive tumors (OR3.2, 95% CI 3.0–3.5). *Conclusions*: Stage I and Stage IV breast cancers represent very different populations of biologic tumor types. This may explain why the incidence of Stage IV cancer has not decreased with screening.

## 1. Introduction

Despite widespread breast cancer screening in the United States, the incidence of de novo Stage IV breast cancer has not decreased. Esserman and colleagues called attention to this irregularity in 2009, showing that localized breast cancer incidence surged with the introduction of disease screening in the 1980s, without a corresponding decrease in distant disease [1]. Works by Bleyer and colleagues have reinforced the conclusion that screening is not meaningfully lowering the incidence of advanced disease [2,3].

The reasons underlying this problem are still unknown. Esserman presciently cited tumor biological factors as likely determinants of disease screenability, calling for the incorporation of such factors into screening and treatment guidelines [1]. Recently, Lannin and Wang showed that small tumors—the majority of which are found on mammography screening—have a distinctly favorable biological profile that dictates an indolent growth pattern [4]. Along similar lines, we hypothesized that de novo Stage IV breast cancer may have a uniquely aggressive biology, granting it growth properties that allow it to escape detection by screening. The purpose of this study was to compare the tumor biology of de novo Stage IV breast cancer with that of small cancers typically detected by screening mammography.

## 2. Methods

### 2.1. Data and Patient Selection

Data for this study were drawn from both the Surveillance, Epidemiology, and End Results database (SEER, November 2017 submission) and the National Cancer Database (NCDB 2015 Participant User File, downloaded 15 December 2017). The original SEER 9 registry data, spanning 1973–2015, were used to analyze long-term population-based incidence trends of the various disease stages. SEER was chosen over NCDB for this analysis, since it contains many more years of incidence data and is population-based and age-adjusted.

NCDB data from 2010 to 2015 were used to explore patient and disease characteristics of Stage IV disease in a large modern population, as well as to compare the tumor biological profiles of Stages I–IV. NCDB was chosen over SEER for this analysis since it contains more robust data on disease characteristics and captures a larger population of breast cancer patients in the United States. The NCDB is a joint project of the Commission on Cancer of the American College of Surgeons and the American Cancer Society. The data used in this study were derived from a de-identified NCDB file. The American College of Surgeons and the Commission on Cancer have not verified and are not responsible for the analytic or statistical methodology employed nor for the conclusions drawn from these data by the investigators.

Included in this study were female patients with invasive breast cancer with known disease stage (in situ and American Joint Committee on Cancer [AJCC] Stage 0 excluded). The analyses of the incidence and patient/disease factors were conducted using targeted statistical methods and software, as outlined below.

### 2.2. Incidence

To identify long-term breast cancer incidence trends by disease stage, the SEER 9 registry data were queried for cases of localized, regional, and distant breast cancer (SEER Historic Stage A variable) from 1973 to 2015. The Historic Stage A variable is traditionally used for analyses prior to 1988, when recoding for AJCC stage was unavailable in SEER. Localized disease includes cancer confined to the breast. Regional disease refers to contiguous organ spread, including regional lymph nodes and the chest wall. Distant disease denotes remote organ metastasis detected at the time of diagnosis. SEER*Stat software (Version 8.3.5, accessed on 20 April 2018) was used to calculate population-based incidence rates and annual percent change (APC). The two-sided *p*-values were set at <0.05.

### 2.3. Patient and Tumor Characteristics

The NCDB was queried for cases from 2010 to 2015 of “NCDB Analytic Stages I–IV,” which uses the AJCC 7th edition pathologic stage classification to collapse sub-stages into their broader designations. Stages I–III cases were consolidated and compared with Stage IV in the univariable analysis of patient characteristics, including race/ethnicity, age, insurance, median household income, and education level, as well as disease characteristics, including histology, tumor size, estrogen receptor (ER) status, progesterone receptor (PR) status, human epidermal growth factor receptor 2 status (Her2), grade, and lymphovascular invasion (LVI). Nodal status was not included in this analysis, as debate exists as to whether lymph node spread marks biological predisposition versus a tumor’s natural history when left untreated. All patients with known disease stage, including those with other missing variables, were included in this analysis, totaling 992,687 patients. Chi-squared testing was used to detect differences in patient and disease variables between Stage IV and non-Stage IV cancer, with statistical significance set at *p* < 0.05.

Five markers of biological activity reported in the NCDB were found to be associated with Stage IV in the univariable analysis: ER, PR, Her2, grade, and LVI. Their values were recombined into 48 possible permutations, generating a spectrum of tumor biology across 740,246 patients for whom these data were available. The rates of Stage IV disease were calculated across the permutation groups, which were then ranked in order of increasing Stage IV percentage. We aimed to cluster these groups into three subtypes of increasing biological aggressiveness, with up to 25% at the extremes, and the remainder intermediately aggressive. After testing multiple Stage IV percentage cut points in sensitivity analyses, we ultimately classified 22.3% of patients as “indolent,” 61.7% of patients as “intermediate,” and 16.0% of patients as “aggressive.” This process of classification is summarized in Table 1.

Frequencies of the above biological subtypes were calculated for all the staged breast cancer cases with known biological data, and their distributions were compared across Stages I–IV. Stage I was divided into tumors measuring 0.1–1.0 cm and 1.1–2.0 cm, in order to compare the biology of tiny tumors almost exclusively found on screening mammography with Stage IV biology. Patients without known tumor size or other demographic or disease variables were excluded from the analysis, resulting in 718,118 patients included. Chi-squared testing was used to detect differences in tumor biology across Stages I–IV, with statistical significance set at *p* < 0.05.

Finally, multivariable logistic regression using backward elimination tested Stage IV cases for significant associations with the demographic variables, including race/ethnicity, age, insurance, and median household income, as well as with disease variables, including histology, size, and biological category. Only the patients with known demographic and disease variables were included, again totaling 718,118 patients. Type I error was set at *p* = 0.05. The analyses were performed using SPSS Statistics software (IBM Version 25, Armonk, NY, USA).

## 3. Results

### 3.1. Breast Cancer Incidence Trends, 1973–2015

Age-adjusted population-based incidence trends for localized, regional, and distant disease are depicted in Figure 1. Based on SEER 9 registry data, the overall incidence of invasive breast cancer increased between 1973 and 2015. The localized disease rate per 100,000 persons increased from 39.0 in 1973 to 85.9 in 2015, generating an APC of 1.20 (95% CI 0.87–1.53). The regional disease rate per 100,000 slightly decreased from 36.9 in 1973 to 34.7 in 2015, with a negative APC of −0.47 (95% CI −0.61 to −0.34). The distant disease rate per 100,000 was lowest but experienced an overall increase from 6.2 in 1973 to 8.7 in 2015, with an APC of 0.46 (95% CI 0.32–0.60).

### 3.2. Demographic and Disease Characteristics of Stage IV Patients

Between 2010 and 2015, there were 992,687 women in the NCDB with staged invasive breast cancer. Of these, 939,903 (94.7%) were Stages I–III, and 52,784 (5.3%) were Stage IV. Univariable analysis of the demographic and tumor characteristics is depicted in Table 2. Analysis of demographic data showed that women with Stage IV disease were more likely to be black, younger than 40 years of age, uninsured or on Medicaid, and living in zip codes where median household income was <$48,000, and where ≥13% of adult residents did not have a high school degree. Strikingly, Stage IV disease affected 8.0% of blacks, 8.8% of women under 30 years of age, and 14.2% of uninsured. Analysis of tumor data showed that women with Stage IV were more likely to have larger tumors, non-ductal or lobular undifferentiated histologies (“Other”), negative ER or PR status, positive Her2 status, LVI, and poorly differentiated grade. Conspicuously, Stage IV affected 19.0% of tumors > 5cm and 10.9% of tumors with undifferentiated histologies.

### 3.3. Tumor Biology Distribution by Stage

As mentioned in Methods, 740,264 patients had known tumor biology characteristics, including hormone receptor and Her2 status, LVI status, and grade. Of this cohort, 718,118 of patients had known tumor size and other demographic and disease characteristics, without missing data. These patients were included in the analysis of the tumor biological subtypes by stage.

Using the tumor biology classification shown in Table 1, a majority of the cohort, 61.7%, had intermediate biology, while 22.7% had indolent biology and 15.6% had aggressive biology. The distribution of biological categories varied tremendously by disease stage (*p* < 0.001 for all stages), as shown in Figure 2. Patients presenting with Stages III and IV disease had similar tumor biology and had over eight times the fraction of aggressive tumors and one-seventh the fraction of indolent tumors as patients with small Stage I tumors ≤1 cm.

### 3.4. Multivariable Analysis of Demographic and Disease Characteristics

In the multivariable logistic regression model, many demographic and disease variables that were associated with Stage IV disease in univariable analysis remained significantly associated, as shown in Table 3. Large tumor size, aggressive biological subtype, and no insurance were the strongest predictors. Tumors with size >5 cm were more than 15 times as likely to predict Stage IV disease as tumors with size ≤2 cm (OR 15.6, 95% CI 14.9–16.5), and tumors with aggressive biology were more than 3 times as likely to present with Stage IV disease as indolent tumors (OR 3.22, 95% CI 2.99–3.47). Race/ethnicity, age, household income, and histology remained significant in the model, but their effects were fairly minor.

## 4. Discussion

Multiple studies have shown that, despite widespread screening programs in the United States, the incidence of Stage IV breast cancer has remained stable or increased over time [1,2,3,5,6]. In this study, we analyzed the most recent and comprehensive population-based SEER data and showed that the incidence of Stage IV disease has indeed been gradually increasing in recent decades.

Esserman and colleagues elegantly delineated this issue in 2009, contrasting the expected stage-based incidence of a theoretically effective disease-screening program with the actual breast cancer incidence trends since the rise of screening in the early 1980s. They highlighted that, while successful screening programs are able to downstage incident cancers over time, breast cancer screening has generated more localized disease diagnoses without congruently diminishing the incidence of advanced cancer [1].

Welch and colleagues pointed out that since the advent of widespread mammography screening, small cancers under 2 cm have increased in incidence over three times more than large tumors over 2 cm have decreased [7]. The clear implication is that not all small cancers are destined to become large cancers, and this leads to overdiagnosis. Lannin and Wang provided an explanation for this by comparing the biology of small and large cancers [4,8]. They found that many breast cancers are small, not because they are detected early, but because they have favorable biology. The current study is an extension of that work and shows that cancers presenting with distant metastases are a distinct subpopulation with a biology much more aggressive than that of the small tumors found by screening mammography. Only a small fraction of the tumors found by screening mammography have the biological profile that puts them at risk for de novo Stage IV disease.

Of course, there are other possible explanations for increasing Stage IV incidence. One contributing factor could be stage migration, the phenomenon whereby the use of high-resolution imaging, including positron emission/computed tomography scans and magnetic resonance imaging, leads to more frequent discovery of distant disease [9,10,11]. It seems unlikely, however, that this would precisely counterbalance a decline in advanced cancer diagnoses that might otherwise be seen from early mammographic detection.

Our data complement the body of literature suggesting that Stage IV cancers arise in unscreened and underprivileged populations, including the very young, the very old, and the disadvantaged with respect to healthcare access and quality. In our analyses, women younger than 40 years and older than 70 years of age had higher rates of Stage IV disease when compared to women aged 40–69 years, who are known to have the highest rates of disease screening [12]. Black women, known to suffer disparate breast cancer outcomes, had significantly higher rates of Stage IV disease than white women [13,14]. Those without health insurance and with Medicaid, as well as from regions in the lowest brackets for income and education—all of which imply low resource settings—also disproportionately presented with Stage IV disease [15,16,17]. These statistics must be interpreted cautiously, however, as NCDB data are not population-based and thus do not differentiate between higher incidence of Stage IV disease versus relatively lower incidence of Stages I–III disease. In fact, Welch and colleagues used population-based SEER data to show that poorer counties with less mammographic screening have lower overall breast cancer incidence but similar Stage IV disease incidence and cancer mortality compared with wealthier counties [18,19].

Even when adjusting for demographic and socioeconomic factors, our data support the concept that Stage IV tumors represent a unique subpopulation and are biologically distinct from the small, indolent tumors usually detected by mammography. Other studies have postulated that breast cancer presenting at an advanced stage may be innately endowed with biologic machinery that promotes swift growth and spread during the 12–24 months interval between mammograms [1,20,21]. Our findings give credence to this theory by highlighting the uniquely aggressive features of Stage IV disease.

In a previous study, Lannin used ER, PR, and grade to stratify patients into three prognostic groups based on breast cancer-specific survival, terming the groups “favorable”, “intermediate”, and “unfavorable”. In this study, we added two additional variables to the model—Her2 and LVI—and used de novo Stage IV disease as the target outcome. These five tumor characteristics were specifically chosen to reflect the biological behavior that predicts distant metastases, thus the group names were changed to “aggressive”, “intermediate”, and “indolent”.

Her2 is a well-known marker for biological aggressiveness and was formerly associated with poor prognosis. With effective targeted therapy, it currently confers a better than average prognosis and yet it is strongly associated with Stage IV disease [22,23,24,25,26]. LVI similarly marks aggressive growth patterns [27,28]. Unfortunately, its status was missing in nearly 20% of the dataset and well over half of Stage IV cases. This may be explained by the fact that many patients with Stage IV disease undergo needle biopsy only, and pathologists are either unable or unmotivated to evaluate for LVI. Despite missing data, we included LVI in our model, as it was highly informative of biology when known. As shown in Table 1, 18 of 19 groups with the highest rate of Stage IV disease presentation were positive for LVI. All 24 groups included in the “aggressive” category were positive for either LVI or Her2. The most biologically unfavorable cancers from the earlier prognostic-based model—triple negative with grades 2 and 3—only cluster to the current “aggressive” category when positive for LVI.

Tumor biology is evolving to become a critical factor in estimating prognosis and guiding treatment. In the AJCC 8th edition disease staging system, anatomical features like tumor size and nodal status are considered insufficient to accurately inform the stage and treatment plan, particularly in the developed world where testing for biomarkers is ubiquitous. The variables used in this study are primitive measures of tumor biology compared to molecular and genomic assays such as OncotypeDx or Mammaprint [4,29,30,31]. However, they are readily available in large datasets like SEER and the NCDB for use in estimating population trends in tumor biology. Future studies will likely elucidate more sophisticated biological mechanisms responsible for aggressive Stage IV tumors. It seems likely that these differing biological characteristics will explain why the incidence of Stage IV breast cancer has not decreased with screening mammography.

## 5. Conclusions

The incidence of de novo Stage IV breast cancer is increasing in the United States despite widespread mammography screening. This is likely related to the differing populations of biologic tumor types that comprise Stage IV tumors versus early-stage tumors commonly found on mammography. Our analysis demonstrates that aggressive tumor biology accounts for nearly 40% of advanced-stage tumors, versus only 5% of tiny early-stage tumors. Conversely, indolent biology is rarely associated with advanced disease. Aggressive biology resulting in insidious growth patterns may explain why the incidence of Stage IV cancer has not decreased with screening.

## Figures and Tables

**Figure 1 cancers-11-00500-f001:**
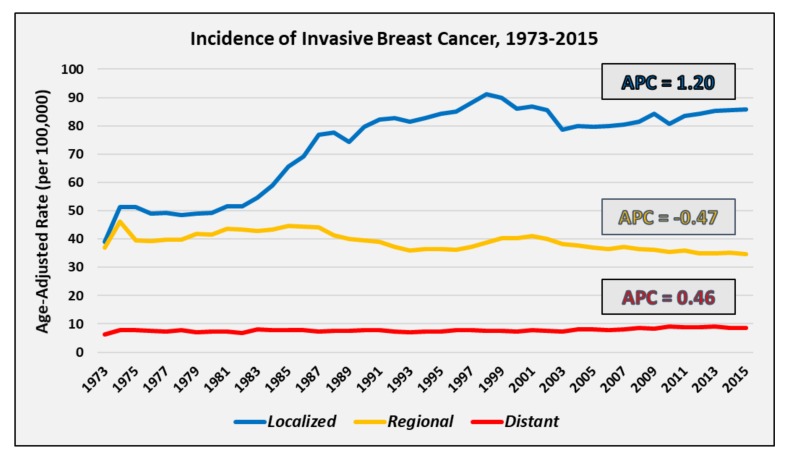
Age-adjusted incidence of localized, regional, and distant breast cancer in women, from 1973 to 2015 (SEER 9). APC: annual percent change.

**Figure 2 cancers-11-00500-f002:**
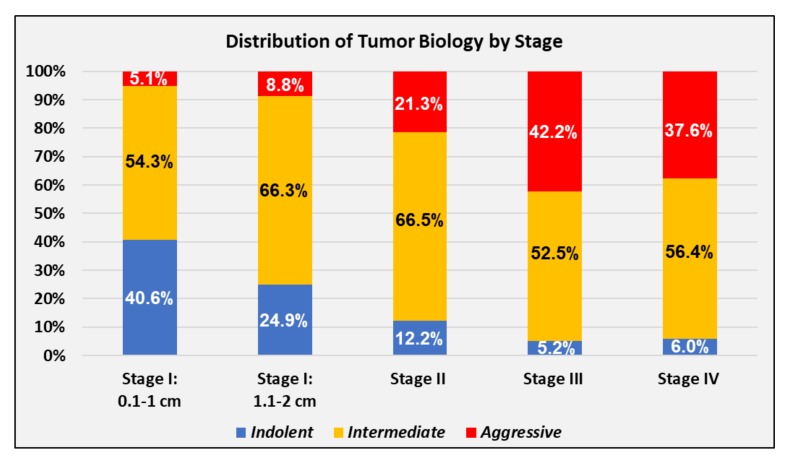
Tumor biology by AJCC 7th edition stage.

**Table 1 cancers-11-00500-t001:** Classification system of breast cancer biological subtypes. ER: estrogen receptor, PR: progesterone receptor, HER2: human epidermal growth factor receptor 2, LVI: lymphovascular invasion.

Group	*n*	ER	PR	HER2	Grade	LVI	Row % with Stage IV	N Subtype	Subtype
1	119	−	+	−	1	−	0	165,150	Indolent
2	7	−	+	−	1	+	0		
3	3	−	+	+	1	+	0		
4	146,900	+	+	−	1	−	0.006		
5	12,409	+	−	−	1	−	0.01		
6	3917	+	+	+	1	−	0.012		
7	1795	−	−	−	1	−	0.013		
8	204,447	+	+	−	2	−	0.014	456,894	Intermediate
9	726	+	−	+	1	−	0.018		
10	691	−	+	−	2	−	0.019		
11	21,860	+	−	−	2	−	0.022		
12	12,597	−	−	−	2	−	0.022		
13	52,160	−	−	−	3	−	0.022		
14	9908	+	+	−	1	+	0.022		
15	18,147	+	+	+	2	−	0.026		
16	48,280	+	+	−	3	−	0.026		
17	3123	−	+	−	3	−	0.026		
18	373	−	−	+	1	−	0.027		
19	754	+	−	−	1	+	0.027		
20	14,045	+	−	−	3	−	0.029		
21	98	+	−	+	1	+	0.031		
22	49,161	+	+	−	2	+	0.031		
23	15,296	+	+	+	3	−	0.033		
24	5228	+	−	+	2	−	0.034		
25	5224	−	−	+	2	−	0.038	118,202	Aggressive
26	14,608	−	−	+	3	−	0.038		
27	6336	+	−	+	3	−	0.042		
28	23	−	+	+	1	−	0.043		
29	949	−	+	+	3	−	0.044		
30	2805	−	−	−	2	+	0.045		
31	411	+	+	+	1	+	0.046		
32	5502	+	+	+	2	+	0.046		
33	318	−	+	+	2	−	0.047		
34	4922	+	−	−	2	+	0.048		
35	28,757	+	+	−	3	+	0.051		
36	17,799	−	−	−	3	+	0.056		
37	6178	+	−	−	3	+	0.059		
38	8808	+	+	+	3	+	0.059		
39	1031	−	+	−	3	+	0.065		
40	147	−	−	−	1	+	0.068		
41	1468	+	−	+	2	+	0.068		
42	473	−	+	+	3	+	0.068		
43	175	−	+	−	2	+	0.069		
44	3168	+	−	+	3	+	0.083		
45	7249	−	−	+	3	+	0.086		
46	1665	−	−	+	2	+	0.091		
47	72	−	−	+	1	+	0.097		
48	114	−	+	+	2	+	0.114		

**Table 2 cancers-11-00500-t002:** Univariable analysis of demographic and disease variables for American Joint Committee on Cancer Stages I–III and Stage IV. HS: high school.

Demographic/Disease Variables	Stages I–III N (Row%)	Stage IV N (Row%)	*p*-Value
**Race/Ethnicity**			<0.001
White	738,416 (95.1%)	38,453 (4.9%)	
Black	103,646 (92.0%)	8988 (8.0%)	
Asian	31,843 (95.6%)	1455 (4.4%)	
Hispanic	50,958 (94.5%)	2983 (5.5%)	
Missing	15,040 (94.3%)	905 (5.7%)	
**Age**			<0.001
<30	4594 (91.2%)	446 (8.8%)	
30–39	36,589 (93.5%)	2560 (6.5%)	
40–49	141,714 (95.6%)	6553 (4.4%)	
50–59	223,898 (94.6%)	12,803 (5.4%)	
60–69	259,993 (94.8%)	14,212 (5.2%)	
≥70	273,115 (94.4%)	16,210 (5.6%)	
**Insurance**			<0.001
None	17,883 (85.8%)	2961 (14.2%)	
Private	467,639 (95.9%)	20,062 (4.1%)	
Medicaid	59,468 (90.2%)	6476 (9.8%)	
Medicare	368,966 (94.5%)	21,609 (5.5%)	
Other Government	9705 (95.9%)	411 (4.1%)	
Unknown	16,242 (92.8%)	1265 (7.2%)	
**Median Household Income**			<0.001
≤$38,000	136,807 (93.2%)	9982 (6.8%)	
$38,000–$47,999	197,306 (94.3%)	11,842 (5.7%)	
$48,000–$62,999	251,060 (94.7%)	13,934 (5.3%)	
≥$63,000	351,943 (95.5%)	16,758 (4.5%)	
Missing	2787 (91.2%)	268 (8.8%)	
**Median Education (No HS Diploma)**			<0.001
≥21%	136,597 (93.3%)	9811 (6.7%)	
13–20.9%	221,484 (94.1%)	13,914 (5.9%)	
7–12.9%	309,839 (94.9%)	16,810 (5.1%)	
<7%	269,536 (95.7%)	12,002 (4.3%)	
Missing	2447 (90.8%)	247 (9.2%)	
**Size (cm)**			<0.001
0.1–2.0	606,385 (98.7%)	8052 (1.3%)	
2.1–5.0	268,290 (93.5%)	18,615 (6.5%)	
>5.0	56,294 (81.0%)	13,216 (19.0%)	
Missing	8934 (40.9%)	12,901 (59.1%)	
**Histology**			<0.001
Ductal	704,671 (95.5%)	33,525 (4.5%)	
Lobular	90,608 (94.2%)	5623 (5.8%)	
Mixed Ductal/ Lobular	49,264 (96.3%)	1912 (3.7%)	
Other	95,360 (89.1%)	11,724 (10.9%)	
**ER**			<0.001
Negative	155,062 (92.9%)	11,810 (7.1%)	
Positive	771,633 (95.5%)	36,720 (4.5%)	
Missing	13,208 (75.6%)	4254 (24.4%)	
**PR**			<0.001
Negative	243,415 (93.0%)	18,354 (7.0%)	
Positive	681,482 (95.8%)	29,586 (4.2%)	
Missing	15,006 (75.6%)	4844 (24.4%)	
**Her2**			<0.001
Negative	763,436 (91.5%)	33,677 (4.2%)	
Positive	119,352 (91.5%)	11,146 (8.5%)	
Missing	57,115 (87.8%)	7961 (12.2%)	
**Grade**			<0.001
Well-Differentiated	212,301 (98.6%)	3107 (1.4%)	
Moderately Differentiated	398,667 (96.1%)	16,333 (3.9%)	
Poorly Differentiated	270,103 (93.4%)	18,953 (6.6%)	
Missing	58,832 (80.3%)	14,391 (19.7%)	
**LVI**			<0.001
Negative	628,236 (98.1%)	12,155 (1.9%)	
Positive	154,951 (95.0%)	8237 (5.0%)	
Missing	156,716 (82.9%)	32,392 (17.1%)	

**Table 3 cancers-11-00500-t003:** Multivariable logistic regression demonstrating the strength of association between Stage IV disease and demographic and disease factors.

**Demographic/Disease Factors**	Stage IV (De Novo) Odds Ratio	95% Confidence Interval
**Race/Ethnicity**		
White	Reference	Reference
Black	1.09	1.04–1.15
Asian	0.78	0.71–0.86
Hispanic	0.71	0.66–0.77
**Age**		
<30	1.38	1.17–1.63
30–39	1.08	0.99–1.18
40–49	0.87	0.81–0.93
50–59	1.07	1.01–1.14
60–69	1.06	1.01–1.12
≥70	Reference	Reference
**Insurance**		
None	Reference	Reference
Private	0.43	0.39–0.46
Medicaid	0.74	0.67–0.81
Medicare	0.51	0.47–0.56
Other Government	0.41	0.34–0.50
Unknown	0.57	0.49–0.68
**Median Household Income**		
≤$38,000	1.12	1.06–1.18
$38,000–$47,999	1.09	1.04–1.14
$48,000–$62,999	1.04	0.99–1.08
≥$63,000	Reference	Reference
**Size (cm)**		
0.1–2.0	Reference	Reference
2.1–5.0	4.40	4.21–4.60
>5.0	15.6	14.9–16.5
**Histology**		
Ductal	Reference	Reference
Lobular	0.83	0.78–0.88
Mixed Ductal/Lobular	0.82	0.75–0.88
Other	0.84	0.79–0.89
**Biological Subtype**		
Indolent	Reference	Reference
Intermediate	2.05	1.91–2.20
Aggressive	3.22	2.99–3.47

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
