# Peer review of "Why Has Breast Cancer Screening Failed to Decrease the Incidence of de Novo Stage IV Disease?"

_cancers, 2019, doi:10.3390/cancers11040500_

Reviewer 1 Report

The authors tried to provide insight into why mammography screening has not decreased the incidence of late stage breast cancer. The methods outlined in the paper do not provide enough detail to fully understand how they chose to focus in on 5 variables or how they assigned the resulting 48 biological subtype permutations to the broader indolent/intermediate/aggressive classification. Thus, it is difficult to know what the paper truly adds to the literature. Assessing which variables are associated with stage at presentation using univariate and multivariate does not seem novel. Maybe if the biological subtype categories were identified using one set of data and then applied to another data set and were shown to track closely with stage then the study might seem more novel. 

Line 40-42. Recently, Lannin and Wang showed that small tumors – the majority of which are found on mammography screening – have a distinctly favorable biological profile that dictates an indolent growth pattern and small phenotype [4]. This is awkwardly written describing small tumors by saying that they have a small phenotype- seems redundant, not informative can “and small phenotype be removed” to simply convey small tumors tend to have slow growth?

Line 81-82. Five markers of biological activity were found to be associated with Stage IV: ER, PR, Her2, grade, and LVI. Why are results from this study presented in the methods section?

Line 81-89. More information is needed to understand how the researchers focused in on the five variables (ER, PR, Her2, grade, and LVI) and then how the assignment of the 48 different permutations were assigned to indolent, intermediate, and aggressive was this subjective, based on modeling technique etc.? 

Line 90 Table 1. It would help to note that the “% Stage IV” is a row percent.

Line 90-95. What relevance do these footnotes have to the results presented in Table 1?

Line 100. Suggest editing to either “SEER 9 registry data” or “SEER 9 registries”

Line 103.  Should there be a negative sign in front of 0.34?

Line 110. Indicates that 992,687 women included. Why does Table 1 only indicate that 740,264 women were included?

Line 121 Table 2 it would be helpful to include sample sizes for “missing/unknown” for all variables.  

Line 123 Indicates 718,118 patients were included – previously stated 992,687 and 740,264. It is very confusing which numbers are correct.

Line 125. “As shown in Table 1, these biological variables were sorted into 48 …” *Note size was indicated in line 123 but is not considered in Table 1. This gets back to the lack of information on how the researchers decided to focus on the 5 variables listed in Table 1 and how the 48 permutations were assigned to the 3 biological subtypes.  

Line 127. Please verify the reported percentages, given the various sample sizes reported it is not clear if they are accurate.

Line 131. Figure 2. Suggest using consistent terminology in Title and footnote: “tumor biology subtype

Line 135-136. On multivariable logistic regression analysis, many of the same demographic, tumor and biological variables remained significantly associated with Stage IV disease, as shown in Table 3.  “Same” as what univariable results from Table 2, given Figure 1 was reported on after Table 2 suggest explicitly stating what you are referring to.

Line 140-141. Race/ethnicity, age, household income, and histology remained significant in the model, but the effects were fairly minor. There is no mention of insurance, and the magnitude of the OR was Private vs. none (0.43) or vis versa (1/0.43=2.33)

Line 178. What is the rationale need to use both SEER and NCDB data?

Author Response

Reviewer 1:

The methods outlined in the paper do not provide enough detail to fully understand how they chose to focus in on 5 variables or how they assigned the resulting 48 biological subtype permutations to the broader indolent/intermediate/aggressive classification. Thus, it is difficult to know what the paper truly adds to the literature. Assessing which variables are associated with stage at presentation using univariate and multivariate does not seem novel. Maybe if the biological subtype categories were identified using one set of data and then applied to another data set and were shown to track closely with stage then the study might seem more novel.  

We have added more detail on the analysis. However, it is important to realize that our goal was not to develop a model to predict patients at risk for Stage IV disease. If it were, it would be quite reasonable to have a training set and a validation set. Instead, our goal was to compare the biological characteristics of tumors likely to be metastatic with those of small mammographically detected tumors. To our knowledge, this has not been done previously. 

Line 40-42. Recently, Lannin and Wang showed that small tumors – the majority of which are found on mammography screening – have a distinctly favorable biological profile that dictates an indolent growth pattern and small phenotype [4]. This is awkwardly written describing small tumors by saying that they have a small phenotype- seems redundant, not informative can “and small phenotype be removed” to simply convey small tumors tend to have slow growth?

Thank you – we removed that phrase. 

Line 81-82. Five markers of biological activity were found to be associated with Stage IV: ER, PR, Her2, grade, and LVI. Why are results from this study presented in the methods section?

Thank you for pointing this out. We included the analysis of biological subtype in the Methods because the resultant analytic variable – Tumor Biology – was one of the measured outcomes of the study and integral to subsequent analyses. The purpose of subsequent analyses was to measure the frequencies of each biological subtype across disease stages, then to assess the association of biological subtype and Stage IV in a multivariable model. Including the a priori analysis of tumor biology in the Methods mirrors Lannin’s previous study that characterized the tumor biology of small breast tumors. 

Line 81-89. More information is needed to understand how the researchers focused in on the five variables (ER, PR, Her2, grade, and LVI) and then how the assignment of the 48 different permutations were assigned to indolent, intermediate, and aggressive was this subjective, based on modeling technique etc.?  

The five selected variables are the only ones in the NCDB that reflect the innate biology of the tumors. In other words, they would not be expected to change as the tumor grew over time. We simply ranked the 48 permutations in order of percent Stage IV disease, then chose two cut points to separate them into three groups. We wanted 15 – 25% at each extreme and the rest in the middle. It turns out that within that range, the results were fairly similar even if the cut points were moved up or down incrementally, which we tested in sensitivity analyses. The cut point we ultimately chose for aggressive disease resulted in the most aggressive 16% of tumors, comprising 38% of Stage IV disease. 

This process has been further clarified in the Methods. 

Line 90 Table 1. It would help to note that the “% Stage IV” is a row percent. 

Done.

Line 90-95. What relevance do these footnotes have to the results presented in Table 1?

Thank you for pointing this out – these lines actually refer to a different analysis, presented in the Results section, Figure 2. The lines have been moved to a new paragraph. 

Line 100. Suggest editing to either “SEER 9 registry data” or “SEER 9 registries”

Done.

Line 103.  Should there be a negative sign in front of 0.34?

Yes, thank you!

Line 110. Indicates that 992,687 women included. Why does Table 1 only indicate that 740,264 women were included?

Table 1 only includes the number of women with known, recorded biology data in the NCDB (ER, PR, Her2, Grade, LVI), which totaled 740,264. Patients with missing data were excluded from that frequency analysis. We have clarified this in the Methods.

Line 121 Table 2 it would be helpful to include sample sizes for “missing/unknown” for all variables.   

Thank you – done. 

Line 123 Indicates 718,118 patients were included – previously stated 992,687 and 740,264. It is very confusing which numbers are correct. 

Thank you for making this point. Explanations for the 3 different sample sizes have been added to the Methods.

Line 125. “As shown in Table 1, these biological variables were sorted into 48 …” *Note size was indicated in line 123 but is not considered in Table 1. This gets back to the lack of information on how the researchers decided to focus on the 5 variables listed in Table 1 and how the 48 permutations were assigned to the 3 biological subtypes.   

We included Stage I tumor size as a condition for the tumor biology frequency analysis because we measured the distribution of biological subtypes across cancer stages, splitting Stage I into 0.1 – 1 cm and 1.1 – 2 cm. Therefore, although size was not one of the variables used to cluster cases into the three biological subtypes (we treated it as a result of tumor biological behavior rather than a driver of it, as was demonstrated in Lannin’s previous study), it was required for this tumor biology frequency analysis. This has been clarified in the Methods and Results.

Line 127. Please verify the reported percentages, given the various sample sizes reported it is not clear if they are accurate. 

Thank you, we have verified the percentages and the sample size used in that analysis are correct. 

Line 131. Figure 2. Suggest using consistent terminology in Title and footnote: “tumor biology subtype

Done.

Line 135-136. On multivariable logistic regression analysis, many of the same demographic, tumor and biological variables remained significantly associated with Stage IV disease, as shown in Table 3.  “Same” as what univariable results from Table 2, given Figure 1 was reported on after Table 2 suggest explicitly stating what you are referring to. 

Thank you for pointing this out - phrasing of this has been edited.

Line 140-141. Race/ethnicity, age, household income, and histology remained significant in the model, but the effects were fairly minor. There is no mention of insurance, and the magnitude of the OR was Private vs. none (0.43) or vis versa (1/0.43=2.33)

We had mentioned insurance status as one of the strong predictive factors of Stage IV early in the paragraph – we’ve now highlighted this point by adding “no insurance” as specifically predictive. 

Line 178. What is the rationale need to use both SEER and NCDB data?

This is an important point that we had not adequately addressed. We have edited the Methods section to include our rationale. 

SEER was chosen over NCDB for the incidence analysis since it contains many more years of incidence data and is population-based and age-adjusted. 

NCDB was chosen over SEER to explore patient and disease characteristics of Stage IV disease since it contains more robust data on disease variables and captures a larger population of breast cancer patients in the United States (about 70% vs 30%).

Reviewer 2 Report

The paper deals with the failure the screening strategy to reduce the incidence of the aggressive stage IV breast cancer at diagnosis, despite an increasingly widespread culture of prevention as a crucial step to fight this type of cancer. The manuscript is well written and falls within the scope of the journal and the goals are very interesting for researchers in the field. However, the reported results seem to be an integration of recent literature data with respect to which there are no particularly significant innovations. However, the software (SEER stat software) used as well as the statistical and multivariable analysis are reasonably worthy of note, so I would suggest to publish the manuscript as a “communication” rather than a research article.

These are some other suggestions:

The conclusions should be improved and extended in a dedicated paragraph, to provide a more comprehensive summary of the entire work.

I also noticed “MD” after authors’ names that should be removed.

Author Response

Reviewer 2

The paper deals with the failure the screening strategy to reduce the incidence of the aggressive stage IV breast cancer at diagnosis, despite an increasingly widespread culture of prevention as a crucial step to fight this type of cancer. The manuscript is well written and falls within the scope of the journal and the goals are very interesting for researchers in the field. However, the reported results seem to be an integration of recent literature data with respect to which there are no particularly significant innovations. However, the software (SEER stat software) used as well as the statistical and multivariable analysis are reasonably worthy of note, so I would suggest to publish the manuscript as a “communication” rather than a research article.

These are some other suggestions:

The conclusions should be improved and extended in a dedicated paragraph, to provide a more comprehensive summary of the entire work.

Thank you, this is an important addition that will help clarify the findings of our work. A Conclusions section has been added.

I also noticed “MD” after authors’ names that should be removed.

Thank you, done.

Round  2

Reviewer 1 Report

The authors' edits have greatly improved the readability of the paper. It would be beneficial if the authors would also incorporate their prior comments to the reviewer into the paper about the selection of the biological category cut points and the sensitivity analyses. This will clarify how these choices were made and the impact on the findings.

“We wanted 15 – 25% at each extreme and the rest in the middle. It turns out that within that range, the results were fairly similar even if the cut points were moved up or down incrementally, which we tested in sensitivity analyses. The cut point we ultimately chose for aggressive disease resulted in the most aggressive 16% of tumors, comprising 38% of Stage IV disease.”

Minor comments:

Line 51 change “SEER 9 registry” to “SEER 9 registry data”

Line 53 .. chosen over NCDB for this analysis since it contains many more years of incidence data, and is

Line 68 change “SEER 9 registry” to “SEER 9 registry data”

Line 129 …no high school degree in >13%. “…and >13% of adult residents with no high school degree.” Please verify measure is no high school degree vs. only a high school degree.  

Line 132 … to have larger tumors, non-ductal or lobular undifferentiated histologies (“Other”), negative ER or PR status,

Author Response

The authors' edits have greatly improved the readability of the paper. It would be beneficial if the authors would also incorporate their prior comments to the reviewer into the paper about the selection of the biological category cut points and the sensitivity analyses. This will clarify how these choices were made and the impact on the findings. 

“We wanted 15 – 25% at each extreme and the rest in the middle. It turns out that within that range, the results were fairly similar even if the cut points were moved up or down incrementally, which we tested in sensitivity analyses. The cut point we ultimately chose for aggressive disease resulted in the most aggressive 16% of tumors, comprising 38% of Stage IV disease.”

Thank you for making this suggestion.  The Methods have been further refined to include the above, with the new paragraph reading:

Five markers of biological activity reported in the NCDB were found to be associated with Stage IV in univariable analysis: ER, PR, Her2, grade, and LVI. Their values were recombined into 48 possible permutations, generating a spectrum of tumor biology across 740,246 patients for whom these data were available. Rates of Stage IV disease were calculated across the permutation groups, which were then ranked in order of increasing Stage IV percent. We aimed to cluster these groups into three subtypes of increasing biological aggressiveness, with up to 25% at the extremes and the remainder intermediately aggressive. After testing multiple Stage IV percentage cut points in sensitivity analyses, we ultimately classified 22.3% of patients as “indolent,” 61.7% of patients as “intermediate,” and 16.0% of patients as “aggressive.” This process of classification is summarized in Table 1.

Minor comments:

Line 51 change “SEER 9 registry” to “SEER 9 registry data”

Done.

Line 53 .. chosen over NCDB for this analysis since it contains many more years of incidence dataand is

Done.

Line 68 change “SEER 9 registry” to “SEER 9 registry data”

Done.

Line 129 …no high school degree in >13%. “…and >13% of adult residents with no high school degree.” Please verify measure is no high school degree vs. only a high school degree.  

Thank you, edited and confirmed. 

Line 132 … to have larger tumors, non-ductal or lobular undifferentiated histologies (“Other”), negative ER or PR status,

Done.